# Cold lenses in the Amundsen Sea: Impacts of sea ice formation on subsurface pH and carbon

Daisy D. Pickup<sup>1</sup>, Dorothee C. E. Bakker<sup>1</sup>, Karen J. Heywood<sup>1</sup>, Francis Glassup<sup>1</sup>, Emily Hammermeister<sup>2,3</sup>, Sharon E. Stammerjohn<sup>4</sup>, Gareth A. Lee<sup>1</sup>, Socratis Loucaides<sup>3</sup>, Bastien Y. Queste<sup>5</sup>, Benjamin G. M. Webber<sup>1</sup>, and Patricia L. Yager<sup>6</sup>

Correspondence: Daisy D. Pickup (D.Pickup@uea.ac.uk)

Abstract. The Amundsen Sea polynya hosts intense sea ice formation, but, due to the presence of relatively warm and salty modified Circumpolar Deep Water, the cold, brine-enriched water is not typically dense enough to sink to the deep ocean. A hydrographic survey of the Dotson Ice Shelf region in the Amundsen Sea using two ocean gliders identified and characterised subsurface lenses containing water with temperatures less than -1.70 °C. These lenses, located at depths between 240 to 500 m, were colder, saltier and denser than the overlying Winter Water (WW) layer. The pH of the lenses was 7.99, lower than WW by 0.02 and the dissolved inorganic carbon concentration was higher in the lenses than WW by approximately 10 μmol kg<sup>-1</sup>. The lenses were associated with a dissolved oxygen concentration greater than surrounding water at the same depth and density due to the cold temperatures increasing O<sub>2</sub> solubility. We hypothesise that these lenses are a product of wintertime surface cooling and brine rejection in areas with intense sea ice formation. They may form in shallow regions, potentially around the Martin Peninsula and Bear Island, where intense upper ocean heat loss occurs, and then spill off into the deeper Dotson-Getz Trough to reach their neutrally-buoyant depth. This is supported by wintertime temperature and salinity observations. This study highlights the importance of shallow parts of shelf seas for generating cold dense water masses in the warm sector of Antarctica. These lenses are widespread in the region of the Dotson-Getz Trough and have the potential to sequester carbon deeper than typical in the region, alongside cooling the water impinging on the Dotson ice shelf base.

#### 15 1 Introduction

Polynyas are referred to as 'sea ice factories', as 10% of Southern Ocean sea ice is formed within major coastal polynyas (Tamura et al., 2008). Coastal polynyas are formed by katabatic winds that push sea ice away from the coast; the newly exposed surface water cools and refreezes before being blown offshore, continually generating open areas for new ice formation during winter (e.g. Golledge et al., 2025). When sea ice forms, it releases salt into the surrounding water through brine rejection. Alongside intense surface heat loss, sea ice formation leads to dense water that will sink once it reaches a sufficient density.

<sup>&</sup>lt;sup>1</sup>Centre for Ocean and Atmospheric Sciences, School of Environmental Sciences, University of East Anglia, Norwich, UK

<sup>&</sup>lt;sup>2</sup>School of Ocean and Earth Sciences, University of Southampton, Southampton, UK

<sup>&</sup>lt;sup>3</sup>Ocean Technology and Engineering, National Oceanography Centre, Southampton, UK

<sup>&</sup>lt;sup>4</sup>Institute of Arctic and Alpine Research, University of Colorado, Boulder, CO, USA

<sup>&</sup>lt;sup>5</sup>Department of Marine Sciences, University of Gothenburg, Gothenburg, Sweden

<sup>&</sup>lt;sup>6</sup>Department of Marine Sciences, University of Georgia, Athens, USA

During sea ice formation, dissolved inorganic carbon (DIC) is rejected alongside salt (Rysgaard et al., 2007), so the remaining dense brine is  $CO_2$ -rich. When this dense brine sinks, it removes DIC from the surface ocean and sequesters carbon in the deep ocean (Rysgaard et al., 2007). This has been observed in tank experiments (Nomura and Yoshikawa-Inoue, 2006) and in situ in the Weddell Sea (Delille et al., 2014), where the partial pressure of  $CO_2$  in brine was oversaturated relative to the atmosphere.

25

The Amundsen Sea polynya is the fourth largest annual cumulative producer of sea ice among Antarctic polynyas (Ohshima et al., 2016; Tamura et al., 2008). This sea ice formation primarily occurs in the Dotson Ice Shelf (DIS) region in shallower water (approximately 300 - 400 m) to the east of the eastern branch of the Dotson-Getz Trough (Ohshima et al., 2016, Fig. 1). In summer, the region is highly stratified with lighter Antarctic Surface Water (AASW) and Winter Water (WW) overlying dense modified Circumpolar Deep Water (mCDW). Typical properties of WW are a minimum conservative temperature ( $\Theta$ ) around -1.8 °C to -1.6 °C and absolute salinities ( $S_A$ ) ranging from 34.05 to 34.15 g kg<sup>-1</sup>. WW is identified as the temperatureminimum layer and is the remnant of the winter mixed layer. The surface water mass, AASW, is warmed by solar heating and freshened by sea ice melt in summer with temperatures above those of the WW layer and  $S_A$  less than 34 g kg<sup>-1</sup>. The mCDW  $(\Theta \text{ greater than } 0 \, ^{\circ}\text{C} \text{ and } \text{S}_A \text{ greater than } 34.8 \text{ g kg}^{-1}) \text{ intrudes onto the continental shelf of the Amundsen Sea from the open$ Southern Ocean, following the troughs to the ice shelves where it contributes to basal melting (Wåhlin et al., 2010, Fig. 1). The presence of this mCDW, along with the insufficiently dense water in the Amundsen Sea polynya, inhibits the full-depth convection seen in other polynyas that leads to the formation of Dense Shelf Water (e.g. Ohshima et al., 2013, 2016; Williams et al., 2008). This has been suggested to make the Amundsen Sea inefficient in contributing to carbon sequestration through the biological and sea ice carbon pumps which aim to transfer dissolved organic and inorganic carbon to the deep ocean. Instead carbon in the Amundsen Sea remains above the mCDW where it is subject to transportation with mid-depth currents (Lee et al., 2017). While the absence of Antarctic Bottom Water formation in the Amundsen Sea is understood, the impact and fate of the dense water generated by persistent sea ice formation in the polynya during winter are not. A recent review on Antarctic polynyas by Golledge et al. (2025) highlighted that many gaps in understanding processes in polynyas - including the role of long-term carbon sequestration in the Amundsen Sea - stem from limited observations.

Studies in the Amundsen Sea polynya have been limited in the past due to its remoteness and lack of access as a result of extensive sea ice. The highest spatial resolution study of the Amundsen Sea polynya to date was the Amundsen Sea Polynya International Research Expedition (ASPIRE) (Yager et al., 2012) in which the stratified water mass structure was described (Randall-Goodwin et al., 2015). From a carbonate chemistry perspective, the main focus of ASPIRE was the surface ocean (Mu et al., 2014) and biological carbon export (Yager et al., 2016). Other campaigns in the region focused on the circulation of water towards and under the ice shelves (e.g Wåhlin et al., 2010; Ha et al., 2014) and meltwater transport from the basally-melting ice shelves (Miles et al., 2016). There is a clear gap in our understanding of subsurface water column dynamics concerning carbonate chemistry in the Amundsen Sea polynya. Currently, no studies investigate the fate of WW or examine the water masses above mCDW beyond the front of DIS.

Higher resolution studies for both physical and biogeochemical properties can be achieved using autonomous underwater vehicles, such as gliders. Unlike most ship-based surveys, the high horizontal spatial resolution of glider observations allows the characterisation of subsurface features on the mesoscale. The Rossby radius in this region is less than 10 km (Chelton

**Figure 1.** Map of the study region within the Amundsen Sea polynya. Arrows indicate the pathway of mCDW following the Dotson-Getz Trough (Wåhlin et al., 2010). Contours show BedMachine v3 bathymetry (Morlighem, 2022). The purple box indicates the boundaries of the glider transect map in Fig. 3c. The dashed blue line represents the area of sea ice production and the dashed green line shows the location of permanent fast ice, both shown in Figure 3 of Ohshima et al. (2016).

et al., 1998), so mesoscale features, such as eddies and meanders, are only a few kilometres in radius. Gliders have been used previously on the West Antarctic Peninsula to detect eddies on the order of 10 km in diameter (Couto et al., 2017). With biogeochemical sensors, including a novel pH sensor, we identify the properties of observed mesoscale features in the Amundsen Sea, and discuss their formation. Supporting data include temperature, salinity, dissolved oxygen ( $O_2$ ), the ratio of oxygen-18 to oxygen-16 ( $\delta^{18}O$ ) and DIC measurements.

#### 2 Methods

60

#### 2.1 Seagliders

Two University of East Anglia Seagliders were used in the southern Amundsen Sea polynya in January and February 2022 (Fig 1). They were deployed from the Research Vessel Icebreaker *Nathaniel B. Palmer* (NBP). This campaign (NBP22-02) was part of the International Thwaites Glacier Collaboration: Thwaites-Amundsen Regional Survey and Network (ITGC: TARSAN) and Accelerating Thwaites Ecosystem Impacts for the Southern Ocean (ARTEMIS) projects. Seaglider 579 (SG579) was deployed approximately 10 km north of DIS on 21 January 2022, completing three circuits of a box (approximately 40 km x 30 km)

in front of DIS, before heading north to 73.41°S 114.24°W, where it was recovered on 21 February 2022. SG579 carried out 247 dives to the seabed, or a maximum depth of 1000 m. Seaglider 510 (SG510) was deployed in open water on 17 January 2022 at 73.81°S 112.61°W and headed south, following the eastern branch of the Dotson-Getz Trough (Fig. 1), in a straight transect to 74.17°S before following the trough northwest to 73.23°S 114.34°W, then south towards Martin Peninsula, crossing the Dotson-Getz Trough, where data collection ceased on 4 February 2022. Prior to communication loss, SG510 transmitted a total of 143 dives to the seabed (up to 1000 m).

Temperature and conductivity sensors (Sea-Bird CT Sail) were integrated on both Seagliders with sampling intervals of 5 seconds which corresponded to a measurement approximately every 0.5 - 1 m (dependent upon glider vertical speed). The raw outputs were processed using the UEA Seaglider Toolbox (Queste et al., 2012) which incorporates corrections for the thermal lag of the un-pumped conductivity cell that can produce artificial salinity spikes following Garau et al. (2011). The mean difference and root mean squared difference between the processed and raw versions of salinity measurements were -0.0011 and 0.0075, respectively for SG579 and -0.0012 and 0.0123, respectively for SG510. In situ temperature and practical salinity were converted to  $\Theta$  and  $S_A$ , using TEOS-10 (McDougall et al., 2010).

An Aanderaa  $O_2$  optode was integrated on SG510 and was lag-corrected using the method of Hahn (2013). The raw measurements were calibrated against an  $O_2$  SBE 43 sensor mounted on the ship's Conductivity-Temperature-Depth (CTD) rosette which itself had been calibrated against discrete water samples for  $O_2$  (section 2.2). To calibrate the glider-measured  $O_2$ , six dives were compared with eight CTD casts that were taken within 3 km of the glider profiles. Only measurements at depths greater than 500 m were compared, so that upper ocean influence of biological activity was excluded. The glider  $O_2$  measurements were on average 7.24  $\mu$ mol L<sup>-1</sup> lower than the CTD  $O_2$  sensor measurements, so this offset was applied. This correction is small relative to the magnitude of the  $O_2$  differences observed between the features discussed in this paper and the surrounding waters. The saturation concentration (DO<sub>sat</sub>) of  $O_2$  was first calculated following (Garcia and Gordon, 1992) using in situ temperature and salinity. This was then converted to percent saturation:  $O_{2sat} = DO/(DO_{sat}) \times 100$ , where DO is the in situ  $O_2$  concentration measured by the optode. For temperature, salinity and dissolved oxygen measurements from the gliders, both ascent and descent profiles are utilised.

A Lab-on-Chip pH sensor developed by the National Oceanography Centre Southampton (Yin et al., 2021), was additionally integrated into SG579. It had a sampling interval of approximately 10 minutes and determined pH on the total hydrogen scale spectrophotometrically (Yin et al., 2021). For pH, only measurements taken during the ascent are reported in this study due to the nature of the glider flight which was programmed to travel relatively slowly during the ascent at a vertical speed of 0.09 m s<sup>-1</sup>, yielding a higher vertical resolution of pH measurements. Final pH values were corrected using in situ temperature, salinity and pressure after calculating the temperature difference between inside the sensor and in situ water following Millero (2007). The pH measurements are reported with an accuracy better than 0.014 and a precision of 0.001 (Yin et al., 2021). The pH sensor stopped recording data on 8 February 2022, after 146 dives of successful data collection.

### 100 2.2 CTD casts and discrete samples

125

130

The CTD sensor on the RVIB Nathaniel B. Palmer is a Seabird SBE 911+ that takes continuous measurements. The CTD was calibrated post-cruise and the corrections were negligible, within the reported accuracy of the sensors (0.001 °C and 0.0003 S m<sup>-1</sup>). Also mounted on the CTD rosette was a dissolved O<sub>2</sub> sensor (SBE43) which was post processed in SeaSoft with time lag and hysteresis corrections applied. It was corrected for an offset of 3.13  $\mu$ mol L<sup>-1</sup> ( $\pm$  0.89  $\mu$ mol L<sup>-1</sup>) based on comparisons between the sensor and Winkler titrated bottle data at depths greater than 500 m (Stammerjohn, 2024). Water samples for 105 DIC and  $\delta^{18}$ O were collected from Niskin bottles mounted on the CTD rosette. Samples for DIC were collected in 500 mL borosilicate bottles following SOP1 from Dickson et al. (2007). They were analysed at the University of Georgia using a Single Operator Multi-parameter Metabolic Analyser attached to a UIC CM5011 CO<sub>2</sub> Coulometer (Yager et al., 2016). Accuracy is based on calibration to two daily certified reference material (CRM) runs, with the average adjustment being  $0.15\pm0.05$  %. Precision based on duplicate runs of CRMs was  $0.56 \mu \text{mol kg}^{-1}$ . Precision based on duplicate samples (repeatability) was 0.90 $\mu$ mol kg<sup>-1</sup>. Water samples for  $\delta^{18}$ O were collected from Niskin bottles mounted on the CTD rosette in 20 mL glass scintillation vials, sealed with poly-seal caps and secured with electrical tape to prevent loosening during storage and shipping. Samples were analysed at the British Geological Survey in the Natural Environment Research Council (NERC) Stable Isotope Facility using the CO<sub>2</sub> equilibration method with an Isoprime 100 mass spectrometer plus Aquaprep device. Isotope measurements were calibrated against internal and international standards including VSMOW2 and VSLAP2. Based on duplicate analysis, analytical reproducibility was better than  $\pm 0.05$  % for all samples.

Measurements for DIC from the ASPIRE campaign are those presented by Yager et al. (2016). The samples were collected between December 2010 and January 2011 and calibrated against CRMs. For further information on collection and analysis see Yager et al. (2016).

# 120 2.3 Stable oxygen isotopes and estimating Meteoric and Sea-Ice Melt Fractions ( $F_{met}$ and $F_{sim}$ )

The depth-discrete oxygen isotope values ( $\delta^{18}$ O), along with CTD-observed absolute salinity ( $S_A$ ), were used to estimate vertical profiles of meteoric and sea-ice melt fractions. To do this, the net freshwater balance is partitioned between the total inputs from all meteoric sources ( $F_{met}$ , precipitation and glacial discharge in the form of melting ice shelves and icebergs) and the input/extraction due to sea ice melt/formation ( $F_{sim}$ ). Following Meredith et al. (2008), we use a three end member mass balance approach (Östlund and Hut, 1984), employing  $S_A$  and  $\delta^{18}$ O as tracers. This approach assumes that the observed  $S_A$  and  $\delta^{18}$ O values in the water column have resulted from a mixture of sea ice melt, meteoric inputs, and the source water mCDW, all three of which have well-separated  $S_A$  and  $\delta^{18}$ O values in pure 'end member' form. Following Hennig et al. (2024), end members for mCDW and meteoric water were empirically determined based on the mCDW–glacial meltwater mixing line, also known as the Gade line (Gade, 1979), using all  $S_A$  and  $\delta^{18}$ O observations below the winter mixed layer (Winter Water core, identified by the temperature minimum); the extrapolated salinity maximum defined the mCDW end member (34.83  $S_A$ , 0.05%  $\delta^{18}$ O), while the zero-salinity intercept defined the meteoric end member (0.0  $S_A$ , -30.32 %  $\delta^{18}$ O). The sea ice end member (7.0  $S_A$ , 2.1%  $\delta^{18}$ O) is based on field observations reported by Randall-Goodwin et al. (2015).

#### 2.4 Seal tags

This study makes use of a combined austral wintertime (April to September) temperature and salinity dataset (3,457 vertical profiles) obtained from 15 seal-borne conductivity-temperature-depth-satellite relayed data loggers (CTD-SRDLs) deployed in the Amundsen Sea in 2014 and 2022. This combined hydrographic dataset was collected by 9 Southern Elephant seals (Mirounga leonina) and 7 Weddell seals (Leptonychotes weddellii). The 2014 hydrographic dataset (1906 profiles) was collected by CTD-SRDLs deployed as part of the UK's Ice Sheet Stability Programme (iSTAR) Ocean2Ice JR294 cruise in February 2014 onboard the RRS James Clark Ross (Heywood et al., 2016). The 2022 hydrographic dataset (9 tagged seals; 1551 profiles) was collected from CTD-SRDLs deployed during the NBP22-02 cruise onboard the RVIB Nathaniel B. Palmer, as part of TARSAN.

Due to the limited transmission bandwidth of the Argos satellite system, CTD-SRDLs only send measurements from 17 to 18 pre-defined depths, transmitting only the deepest ascent in every 4 to 6 hour window (Boehme et al., 2009). This study accesses the seal-borne CTD-SRDLs profiles via the Marine Mammals Exploring the Oceans Pole to Pole (MEOP) server which applies standard delayed-mode quality control and vertically interpolates profiles to 1 m (for details see: https://www.meop.net/database/data-processing-and-validation.html). Before use in this study, profiles were further calibrated against concurrent summertime ship-based hydrography and historical (1994-2020) profiles from the MIPkit-A dataset (Nakayama et al., 2024). Calibration offsets for temperature and salinity were calculated manually for each CTD-SRDL following comparisons with regional observations of mCDW properties, i.e. Θ greater than 0 °C at depths greater than 700 m. CTD-SRDL GPS coordinates were adjusted using standard repositioning algorithms derived using a Kalman smoother (Savidge et al., 2023; Lopez et al., 2015).

#### 3 Results

#### 3.1 Location of lenses

Analysis of the water masses in temperature-salinity space reveals two distinct salinity ranges where  $\Theta$  is less than -1.60 °C (Fig. 2). The more saline water mass (with  $S_A$  greater than 34.2 g kg<sup>-1</sup>) is denser with temperatures closer to the freezing line (as calculated using TEOS-10 (McDougall et al., 2010)). The less saline water mass (with  $S_A$  less than 34.2 g kg<sup>-1</sup>) is the WW layer that lies immediately below the AASW (Fig. 3a and b), with an  $S_A$  between 34.05 and 34.15 g kg<sup>-1</sup> (Fig. 2). The colder and more saline water appears as isolated patches within the subsurface water column, below the WW layer (Fig. 3a and b). Such localised subsurface bodies of water with distinct physical and chemical properties are commonly referred to as lenses, for example meddies found in the Northeast Atlantic emanating from the Mediterranean (McDowell and Rossby, 1978; Armi et al., 1988). For the purpose of this study, this terminology will be adopted to describe these isolated, cold features. The lenses are located in the stratified layer at approximate depths of 200 to 500 m between the warmer of the two temperature minima of WW and the warmer deeper mCDW, the deepest water mass (Fig. 3a and b). The mCDW has  $\Theta$  above 0.35 °C and  $S_A$  greater than 34.72 g kg<sup>-1</sup> (Fig. 2), which agrees well with previous in situ observations in the Dotson region (Randall-Goodwin et al., 2015; Miles et al., 2016; Yang et al., 2022).

In total, ten lenses are identified from the glider transects (Fig. 3). The lenses vary in apparent size, with the largest horizontal extent, as captured by the glider, reaching a maximum of 25 km and the smallest being approximately 5 km. All of the lenses are in locations where the total water column depth is greater than 500 m and, with the exception of C and F, all of the lenses are located in water with depths greater than 600 m. Two of the lenses (A and B) are located at approximately 73.9°S, within 25 km from each other and were captured 8 days apart (Fig. 3a and b). We argue that A and B are the same lens and the differences in size could be due to the glider crossing either the centre of the lens, or an edge. This is supported by the comparison of the core lens properties to the extremities. The  $\Theta$  of lens A increases towards the edges to -1.60 °C, similar to the core (coldest) value of lens B. Thus, the core  $\Theta$  of lens A is better representative of the properties of the lenses than that of B, which is also supported by the properties of the other identified lenses shown in Table 1.

As the gliders only surveyed the eastern branch of the Dotson-Getz Trough (Fig. 3c), CTD casts covering a wider area (Fig. 4) were also investigated for the presence of lenses, allowing further biogeochemical analysis of the lens water masses. The CTD casts with lenses were identified as water masses with Θ less than -1.70 °C at a depth greater than 200 m. These thresholds were chosen as all of the glider lens water masses had temperatures less than -1.70 °C (Table 1), apart from lens B which will be explored later. The CTD casts with lenses detected were: 8, 10, 16, 18, 231, 279 and 294 (Fig. 4). The casts were located in similar regions to the lenses identified by the gliders, in the proximity of the eastern branch of the Dotson-Getz Trough (Fig. 4). No lenses were found either near the Dotson or Getz ice shelf fronts, or along the shallow bank north of Bear Island where there were CTD casts. This suggests that the formation of lenses observed in this campaign was through a mechanism local to the area around the eastern branch of the Dotson-Getz Trough. The CTD casts show that the lenses also appear further north in the trough (between 73.0 and 73.5 °S), where the two Seagliders did not survey. Despite being constrained to the trough area, the location of the lenses varies, from the deepest part where the water depth is 900 m to the shallower trough edge with a water depth of 550 m (Figs. 3c and 4).

Three lenses (D, J and CTD 279) are located at approximately 73.43°S, within 5 km from each other, within a 19 day period (Fig. 3a and 4). This suggests that it is the same lens, although there is a significant difference in size between D and J (approximately 20 km). Lens D was surveyed by SG579 just before recovery and therefore the coverage of lens D was cut short. Additionally, the two gliders were not following the same path, which means different parts of the lens were likely captured. The lens identified in CTD 18 is in approximately the same location as lens E, three hours later. Since the CTD is a single profile, it is not possible to determine the size or shape of the lens captured. The occurrence of these repeatedly surveyed lenses indicates that lenses are not short-lived features on the timescale of hours or days. The comparison between lenses in similar locations suggests that the size variation between surveys of the same lenses is an artefact of how the lens was surveyed.

## 3.2 Core properties of the lenses

190

The end members of WW and mCDW and the core properties of each identified lens are shown in Table 1. These have been determined using both the glider and CTD measurements, where measurements of a variable exist for each. For mCDW and WW, the associated biogeochemical properties are those that align with the  $\Theta$  and  $S_A$  end members. For the lenses the core is defined as the temperature minimum. This was checked using the standard deviations for the properties in all of the lenses

**Figure 2.** Temperature-salinity diagram for measurements in the study area from the gliders and CTD casts coloured by depth. Local water masses are labelled and the freezing line was calculated using TEOS-10 (McDougall et al., 2010). End members are shown with a cross and grey dashed lines indicate water mass mixing lines. Black sloping lines depict potential density (kg m<sup>-3</sup>). The black dashed line depicts the location and properties of the lenses.

(Table A1) which were all small. With the exception of lens B, all of the identified lenses have a core  $\Theta$  less than -1.70 °C. The mean core  $\Theta$  of all of the lenses (excluding B) is -1.80 °C which is 0.05 °C lower than the end member of the overlying WW and 2.51 °C lower than the mCDW end member. The core of the lenses are also more saline and denser than WW, with a maximum difference of 0.23 g kg<sup>-1</sup> and 0.16 kg m<sup>-3</sup>, respectively. Conversely, mCDW is up to 0.51 g kg<sup>-1</sup> saltier and 0.30 kg m<sup>-3</sup> denser than the core of the lenses.

Measurements of pH show that the core of lens A had a pH 0.01 less than overlying WW (Table 1) and greater than deeper mCDW by 0.08. The lowest pH is found in mCDW as it is an old water mass that has remained isolated from the surface for an extended period of time, so DIC has accumulated in it through remineralisation. In WW, pH is higher than in mCDW as there has been recent surface exchange with the atmosphere, during a recent winter. As the lenses exhibit a pH more closely resembling that of WW it could be assumed that they have been at the ocean surface at a similar time. Concentrations of  $O_2$  in the core of the lenses range between 278 - 293  $\mu$ mol  $L^{-1}$  (Table 1) with a mean of 291  $\pm$  2. They are similar to, or slightly higher than,  $O_2$  concentrations in WW (281  $\mu$ mol  $L^{-1}$ ) and much higher than in mCDW (195  $\mu$ mol  $L^{-1}$ ) as  $O_2$  has been consumed in mCDW through remineralisation and there has been no exchange with the surface to replenish depleted  $O_2$ . As with pH, the similarity between  $O_2$  in the lenses and WW suggests recent interaction with the atmosphere. The slight increase

205

Figure 3. Lenses identified in glider transects.  $\Theta$  is shown for depth and distance travelled by a) SG579 and b) SG510. Black contours show potential density (kg m<sup>-3</sup>). Cold lenses (A-J) are outlined in white dashes. c) The location of the glider transects (blue for SG579 and black for SG510) and the lenses (green). The purple crosses depict where each respective glider transect starts. Bathymetry is from BedMachine v3 (Morlighem, 2022) and isobaths are plotted in white.

from WW to the lenses could be a result of colder water being able to absorb more  $O_2$  from the atmosphere during ventilation, however, the  $O_2$  saturation is only 2 - 3% higher than WW in more than half of the lenses that have  $O_2$  measurements.

The differences between the lenses captured by both CTD casts and gliders (i.e. D, J and 279 as well as 18 and E) could be due to the fact the CTD casts are a single profile and do not extend horizontally. The temperature minimum in each single CTD profile is considered the core temperature of the lens, whereas the glider profiles reveal that the lenses can be as large as 25 km horizontally. The glider profiles also show that the lens properties can vary between the core and extremities. This highlights a limitation of lower resolution CTD sampling.

#### 3.3 Comparison with surrounding water

220

Lenses observed in each CTD cast are located at different depths, with the  $\Theta$  minimum found within the range of 240 and 405 m (Fig. 5). The varying profile shapes within the lenses possibly reflect the same feature observed by the glider transects, where the difference in lens shape and size depends on which part of the lens was captured by the single profile. Three of the CTDs with lenses (Fig. 4) were additionally sampled for  $\delta^{18}$ O and DIC as shown in Fig. 5d and e. For CTD 8 the  $\delta^{18}$ O decreases

**Table 1.** Properties of the main water masses and lenses identified in the Amundsen Sea Polynya determined from glider and CTD measurements. For WW this was determined by identifying the temperature minimum and for mCDW the temperature maxima and then selecting the associated values for the other properties, or those closest depending on sampling resolution. The core properties of the cold lenses in the Dotson Ice Shelf region from the Seagliders are labelled A-J corresponding to Fig. 3 and CTD casts numbers correspond to Fig. 4. The core lens properties were identified by finding values associated with the temperature minimum, or closest value.

|              | Θ (°C) | $S_A (g kg^{-1})$ | $\sigma\theta({\rm kg~m^{-3}})$ | pН   | $O_2$ ( $\mu$ mol $L^{-1}$ ) | ${\rm DIC}\ (\mu{\rm mol}\ {\rm kg}^{-1})$ | $\delta^{18}{\rm O}\left(\%e\right)$ | $O_{2sat}$ (%) |
|--------------|--------|-------------------|---------------------------------|------|------------------------------|--------------------------------------------|--------------------------------------|----------------|
| Water masses |        |                   |                                 |      |                              |                                            |                                      |                |
| WW           | -1.75  | 34.09             | 27.34                           | 8.00 | 281                          | 2212                                       | -0.47                                | 74             |
| mCDW         | 0.71   | 34.78             | 27.76                           | 7.91 | 195                          | 2258                                       | 0.01                                 | 55             |
| Lenses       |        |                   |                                 |      |                              |                                            |                                      |                |
| A            | -1.85  | 34.29             | 27.48                           | 7.99 |                              |                                            |                                      |                |
| В            | -1.62  | 34.25             | 27.44                           |      |                              |                                            |                                      |                |
| C            | -1.84  | 34.29             | 27.48                           |      |                              |                                            |                                      |                |
| D            | -1.76  | 34.26             | 27.45                           |      |                              |                                            |                                      |                |
| E            | -1.85  | 34.29             | 27.48                           |      | 289                          |                                            |                                      | 76             |
| F            | -1.78  | 34.31             | 27.49                           |      | 281                          |                                            |                                      | 74             |
| G            | -1.80  | 34.27             | 27.46                           |      | 281                          |                                            |                                      | 74             |
| Н            | -1.81  | 34.31             | 27.49                           |      | 282                          |                                            |                                      | 74             |
| I            | -1.80  | 34.28             | 27.47                           |      | 282                          |                                            |                                      | 74             |
| J            | -1.82  | 34.28             | 27.47                           |      | 278                          |                                            |                                      | 74             |
| CTD 8        | -1.81  | 34.32             | 27.49                           |      | 293                          | 2222                                       | -0.50                                | 77             |
| CTD 10       | -1.81  | 34.31             | 27.49                           |      | 292                          |                                            |                                      | 77             |
| CTD 16       | -1.82  | 34.31             | 27.49                           |      | 293                          |                                            |                                      | 77             |
| CTD 18       | -1.82  | 34.31             | 27.48                           |      | 291                          | 2221                                       | -0.51                                | 77             |
| CTD 231      | -1.77  | 34.32             | 27.49                           |      | 291                          |                                            |                                      | 77             |
| CTD 279      | -1.73  | 34.27             | 27.45                           |      | 288                          | 2222                                       | -0.52                                | 76             |
| CTD 294      | -1.79  | 34.29             | 27.47                           |      | 291                          |                                            |                                      | 77             |

from -0.45 % at 200 m in WW to -0.50 % at 300 m within the lens. For CTD 18 the decrease in  $\delta^{18}$ O is from -0.40 % at 200 m (WW) to -0.48 % within the lens at 400 m. For CTD 279 the decrease is from -0.47 % at 245 m (WW) to -0.55 % at 345 m, just below the lens. In the WW (depths between 100 - 200 m)  $\delta^{18}$ O is approximately 0.5 % more depleted than in mCDW (depths greater than 450 m). Although there are only seven data points of  $\delta^{18}$ O within the lenses and sample distribution with depth is sparse, these results suggest that the  $\delta^{18}$ O values of the lens cores are more depleted than those of WW (Table 1).

The effect of sea ice formation and melt on  $\delta^{18}$ O is small compared with meteoric inputs, whereas the effect on salinity is large. The derived freshwater contributions of meteoric input and sea ice formation, calculated from  $\delta^{18}$ O, allow us to identify the processes affecting the lenses. The freshwater fractions (Fig. 6) reveal a negative sea ice contribution to the lenses which

Figure 4. Location of CTD casts during NBP22-02 ship-based campaign. CTD casts that did not capture a lens are shown as black crosses. CTD casts with a lens are shown as blue markers and identified with their cast numbers, with blue circles for profiles with sampling for DIC and  $\delta^{18}$ O and blue crosses for those without such sampling. The colours of CTDs 8, 18 and 279 correspond to those in Fig. 5. Comparison CTD 194 (filled white circle) for temperature, salinity,  $O_2$  and  $\delta^{18}$ O is at the same location as CTD 18. Grey markers show locations of comparison CTD casts from ASPIRE for DIC. Purple lines indicate the two glider tracks.

indicates net sea ice formation, greatest for CTD 18 and least for CTD 8. There is no  $\delta^{18}$ O measurement at the depth within the lens from CTD 279, but the measurement just below the lens also shows a negative sea ice melt contribution. Water above the lenses, especially close to the surface, has a positive sea ice meltwater contribution, indicating net sea ice melt which has occurred locally in the spring.

235

240

DIC concentrations in the lenses are slightly higher than in WW, by up to  $10~\mu mol~kg^{-1}$  (Table 1). Similar to pH and  $O_2$ , mCDW has the highest DIC concentration (2261  $\mu mol~kg^{-1}$ ) as it has spent the longest time isolated from the atmosphere. Additionally, DIC accumulates in the water column due to the breakdown of organic matter which produces DIC.

The CTD casts with a lens are compared with a CTD profile where a lens was not observed (Fig. 5). For temperature, salinity,  $O_2$  and  $\delta^{18}O$ , the comparison CTD cast (CTD 194) was in the same location as the lens CTD cast (CTD 18), 20 days later (Fig. 4). No additional CTD casts were sampled for DIC in the region of the Dotson Trough during the TARSAN-ARTEMIS campaign. Instead, two CTD casts were sourced for comparison from the ASPIRE campaign in December 2010 - January 2011 (Yager et al., 2012), within 25 km of CTD 18 (Fig. 4). Three consecutive SG579 dives immediately after the glider exited lens A were selected for a comparison of pH profiles in the lens with pH profiles in surrounding water.

**Figure 5.** Depth profiles with lenses in blue and a comparison profile(s) in grey. a - e) data from the 3 CTD casts with a lens and biogeochemical sampling. f) data from SG579, lens A. See Fig. 4 for location of the comparison profiles. Horizontal lines depict the location of the lens in the water column and the colour of this line corresponds to the CTD cast/glider profile.

Temperature and salinity are lower in the lenses than in nearby water at the same depth (Fig. 5) by up to 1.50 °C and 0.16 g kg<sup>-1</sup>, respectively. The dissolved  $O_2$  concentration and  $O_2$  saturation is higher in the lenses than in surrounding water (by approximately 50  $\mu$ mol L<sup>-1</sup> and 11%, respectively) indicating the ability of cold water to dissolve more  $O_2$ . The  $\delta^{18}O$  in the lenses is approximately 0.18 % lower than surrounding water. The lens has DIC concentrations very similar to one of the ASPIRE CTD casts (CTD 9) but approximately 18  $\mu$ mol kg<sup>-1</sup> lower than the other (CTD 72, collected about 2 weeks later in

Figure 6. Vertical profiles of freshwater fractions from meteoric (blue) and sea ice melt (green) sources derived from  $\delta^{18}$ O and S<sub>A</sub> measurements for the three CTD casts with lenses. The horizontal blue dashed lines indicate the locations of the lenses with colours corresponding to CTD profile following Figs 4 and 5.

the season and over a deeper part of the trough). The lower DIC concentration is consistent with pH which is 0.02 higher in the lens than in surrounding water at the same depth.

Evidence of deep convection from the surface to depths less than 400 m or to the seabed (if shallower than 400 m), as indicated by seal profiles, was observed on the shallow shelf around Martin Peninsula and to the east of the Dotson-Getz Trough (Fig. 7a), plus very occasionally in deeper water across the trough. Temperatures within these profiles reach a minimum of -1.85 °C at depths of approximately 300 - 400 m (Fig. 7b). These temperatures are associated with  $S_A$  greater than 34.25 g kg<sup>-1</sup>, consistent with the properties of the lenses.

## 4 Discussion

255

The low temperature, relatively high salinity and relatively high density of the lenses compared with WW suggest that the lenses are formed by a mechanism similar to that of WW, but more intense. The 2014 seal tag profiles discussed by Zheng et al. (2025) in the area north of Bear Peninsula revealed intense heat loss in the surface ocean alongside deepening of the mixed layer and salinification from brine rejection. Heat and salt budgets suggested a sea ice formation rate of approximately 3 cm day<sup>-1</sup> (Zheng et al., 2025). Therefore, we are confident that sea ice formation and cooling play a role in lens development. The water surrounding the lenses at the same depth is a mixture of local WW and mCDW. This mixing would lead to higher temperatures

Figure 7. Seal tag CTD profiles in austral autumn/winter from 2014 and 2022. (a) Locations of seal tag profiles from 2014 and 2022, coloured by month. Circles outlined in black are those that meet criteria for deep convection from the surface to depths less than 400 m or to the sea bed if shallower, and have  $\Theta$  less than -1.7°C,  $S_A$  between 34.25 and 34.35 g kg<sup>-1</sup>, and potential density anomaly between 27.45 and 27.55 kg m<sup>-3</sup>. The criteria for deep convection was defined by a surface to deepest value difference of  $\Theta$  less than 0.05 °C and  $S_A$  less than 0.0250 kg<sup>-1</sup>. (b)  $\Theta$  and  $S_A$  of seal tag profiles within region, coloured points in the foreground correspond to the profiles outlined in black in a). Curved dotted lines represent potential density (kg m<sup>-3</sup>) contours and the freezing line is the black dashed line.

**Figure 8.** Schematic depicting (left hand side) the two hypothesised formation mechanisms of the lenses and their properties. The numbers reference the formation mechanism with (1) formation in shallow water due to sea ice formation and spilling into deeper water and (2) local chimneys of convective mixing. The right hand side depicts the properties of the overlying WW relative to the lenses.

and salinities than WW due to the mCDW properties. The mCDW contribution would also have a low O<sub>2</sub> concentration, less depleted δ<sup>18</sup>O values, high DIC concentration and low pH content. The properties of the lenses indicate that they have not mixed with mCDW. The notable difference between the properties of the lenses and nearby water indicates that the lenses may not have formed locally. We hypothesise that dense water in the lenses is produced during winter by cooling and sea ice formation. Increased cooling would increase brine rejection, through sea ice formation, and both would increase density.

The lens properties sit just warmer than the freezing line, much closer to it than WW (Fig. 2). The greater density mCDW prevents the lenses from sinking to the bottom of the water column and contributing to overturning, unlike polynyas in the cold sectors of Antarctica. Water as cold as our observations at these depths has been seen in similar locations by Randall-Goodwin et al. (2015) in 2010/2011 and Miles et al. (2016) in 2014. This suggests that the lenses are recurring features, and supports the theory that they could form annually through sea ice formation. The findings from this study suggest that the formation mechanism is particularly prevalent around the Dotson region, in comparison to nearby Getz which was also surveyed. The

higher DIC concentration within the lenses than the overlying WW also suggests the importance of sea ice formation as DIC is rejected alongside salt during the freezing of seawater (Rysgaard et al., 2011; Delille et al., 2014). The increased O<sub>2</sub> and DIC in the lenses relative to WW could indicate that stronger surface cooling was present, increasing the solubility of O<sub>2</sub> and CO<sub>2</sub> and leading to more uptake and a higher O<sub>2</sub> saturation. Further support for the role of sea ice formation in lens development comes from the estimated freshwater fractions. The negative contribution of sea ice melt and subsequent indication of sea ice formation at only the depth of the lenses indicates sea ice formation plays a role in their creation. The values of sea ice melt in WW are less negative than in the lenses, despite sea ice formation also taking place in WW. Properties of WW vary depending on the circumstances when it formed (not necessarily the previous winter), such as the extent of sea ice formation, wind strength and direction, and net surface heat loss.

We propose two possible mechanisms for lens formation (Fig. 8). The first is that they are formed in shallow waters where there is strong surface heat loss and little ability to gain heat by mixing warm water from below. We suggest that this could partly occur nearby in the 300 - 400 m water to the east of the Dotson-Getz Trough, where studies have highlighted a key area of sea ice formation (Ohshima et al., 2016, and highlighted in Fig. 1) due to the presence of the persistent polynya. Winter sea ice concentrations six months prior to our field campaign show a persistent wintertime polynya there, with intense sea ice formation (Fig. 9). This area also exhibits seal tag profiles in winter that show deep convection (Fig. 7a). Intense heat loss could also occur in the shallow region surrounding Martin Peninsula to the west of the Dotson-Getz Trough, where there is also sea ice production (Ohshima et al., 2016). It is possible that the lenses then spill off the shallow regions and follow isopycnals to become neutrally buoyant at approximately 400 m depth in deeper areas, in a similar mechanism to meddies (McDowell and Rossby, 1978), and sit above the dense mCDW. The WW above the lenses has different properties as it may be formed locally in deeper regions, with less intense cooling and sea ice formation.

The second postulated mechanism is local formation through intense deep convection in winter, in a similar form to convective chimneys observed in the Arctic (Wadhams et al., 2002). Deep convection requires intense surface cooling and brine rejection. We suggest that as the surface ocean warms and re-stratifies in spring, the chimneys break down (Marshall and Schott, 1999) and the water formed by this deep convection remains at depth. As these are localised features, surrounding water at the same depth has different properties. Further hydrographic profiles, particularly during autumn and winter, both in the trough and on the surrounding shallow regions, would help to determine whether one or both of the hypothesised mechanisms commonly occur. For example, deployments of profiling/grounding biogeochemical Argo floats would enable observations under the sea ice in winter.

Whilst previous studies have seen water masses with these lens properties in the Amundsen Sea region during summer (Randall-Goodwin et al., 2015), this is the first time that these lenses have been investigated. There remains unanswered questions that merit further research, such as the impacts of the lenses. Steiger et al. (2021) found that the deep WW layers located at the nearby Getz Ice Shelf reduce the ocean heat transport into the Getz Ice Shelf cavity. In a similar process, the lenses reduce heat content and could act to reduce the transport of heat under DIS. Another potential effect is that if the lenses at the depth of the base of the DIS were to advect under it, they may act as a barrier to basal melting. We did not observe lenses at the ice front or in the cavity, but high resolution measurements of water masses and currents beneath the ice shelf would be

**Figure 9.** Sea ice cover from AMSR2 (%) for 18 July 2021 in the southern Amundsen Sea. The light blue areas within the black lines have less than 15% sea ice cover. White areas indicate a lack of data.

needed to confirm this. The fate of the lenses is not determined in this study. Whilst the Amundsen Sea has been identified as a region of minimal carbon sequestration due to the presence of mCDW preventing the formation of deep water masses with a high carbon content (Lee et al., 2017), the lenses do provide a mechanism for transporting carbon deeper than typical WW can achieve (approximately 200 m). Further studies are required to understand the fate of this carbon in the lenses and its ultimate fate.

Surveys carried out by the autonomous underwater gliders allowed the identification of the size, shape and properties of the lenses. In our study, glider profiles were spaced 1–2 km apart, comparable to the spacing used by Couto et al. (2017), who successfully resolved eddy features with scales of 10 km. Given that the lenses we observed were up to 2–3 times larger than those eddies, we are confident that our glider resolution was sufficient to capture the full and representative distribution of lenses present in the study area. This suggests that surveys in the Amundsen Sea with novel, high resolution sensors can improve our understanding of subsurface features. There was no signature of the lenses in dynamic height, calculated using in situ  $\Theta$ ,  $S_A$ , pressure and a reference pressure of 500 m, indicating that the lenses would not be identifiable using satellite altimetry.

#### 5 Conclusions

In the southern Amundsen Sea, isolated subsurface lenses at depths of 240 to 500 m have been identified in the area of the eastern branch of the Dotson-Getz Trough. To our knowledge, this is the first study to analyse these lenses in detail. The lenses

are colder, saltier and denser than overlying WW and are associated with a higher DIC concentration, lower pH and an enhanced fraction of net sea-ice formation (as determined from  $\delta^{18}$ O-derived freshwater fractions). We hypothesis that the lenses are a denser, colder variety of WW, formed under different conditions. Evidence from the lens features, the estimated sea ice melt and wintertime seal tag profiles suggests that the lenses are formed by a combination of intense cooling and brine rejection. These processes are possibly aligned with regions of strong sea ice production and/or shallower waters, which cause the lenses to have different properties from overlying WW. The lenses sit above mCDW as they are not dense enough to sink below it. Further studies are needed to understand how the lenses move and lose their characteristics, how their volume and properties vary, where else in the region they may be located and where they form. Depending on their movement, the lenses could have implications for the transfer of heat underneath regional ice shelves, such as DIS, and could act as a buffer to basal melting. They may also provide a mechanism for sequestering carbon to deeper depths than typical WW can achieve. The identification and characterisation of these lenses highlight the ability of autonomous vehicles to capture features that are not easily detectable using single ship-based profiles. The detection of these lenses not only deepens our understanding of regional oceanographic processes, but also demonstrates the role of autonomous observations in revealing smaller, yet potentially impactful, features.

Data availability. The glider datasets are publicly available on the British Oceanographic Database Centre (doi:10.5285/35b7832c-2432-fb41-e063-7086abc0c610). The pH dataset from SG579 is available separately at: doi:10.5285/33d1f143-47b5-a7d1-e063-7086abc03081. The 2022 CTD measurements are publicly available on the United States Antarctic Program Data Center: doi.org/10.15784/601785. The seal data are available at MEOP: https://www.meop.net/database/meop-databases/index.html.

Auxiliary data utilised in figures can be found at data.seaice.uni-bremen.de (sea ice) and https://worldview.earthdata.nasa.gov/ (satellite imagery). BedMachinev3 is available from the National Snow and Ice Centre Database (https://nsidc.org/data/nsidc-0756/versions/3).

## Appendix A

350

|         | Θ (°C) | $S_A (g kg^{-1})$ | $\sigma\theta~({\rm kg~m^{-3}})$ | pН    | $O_2 \ (\mu mol \ L^{-1})$ |
|---------|--------|-------------------|----------------------------------|-------|----------------------------|
| A       | 0.031  | 0.008             | 0.006                            | 0.006 |                            |
| В       | 0.024  | 0.014             | 0.011                            |       |                            |
| C       | 0.006  | 0.008             | 0.006                            |       |                            |
| D       | 0.015  | 0.009             | 0.007                            |       |                            |
| E       | 0.032  | 0.007             | 0.006                            |       | 3.2                        |
| F       | 0.021  | 0.007             | 0.005                            |       | 2.9                        |
| G       | 0.024  | 0.009             | 0.007                            |       | 2.2                        |
| Н       | 0.019  | 0.008             | 0.007                            |       | 1.0                        |
| I       | 0.018  | 0.010             | 0.008                            |       | 1.5                        |
| J       | 0.025  | 0.004             | 0.003                            |       | 1.4                        |
| CTD 8   | 0.006  | 0.008             | 0.006                            |       | 0.7                        |
| CTD 10  | 0.005  | 0.004             | 0.004                            |       | 0.4                        |
| CTD 16  | 0.009  | 0.011             | 0.009                            |       | 1.3                        |
| CTD 18  | 0.021  | 0.027             | 0.022                            |       | 3.7                        |
| CTD 231 | 0.041  | 0.004             | 0.002                            |       | 3.3                        |
| CTD 279 | 0.009  | 0.006             | 0.005                            |       | 1.2                        |
| CTD 294 | 0.011  | 0.006             | 0.005                            |       | 1.8                        |

**Table A1.** Standard deviations of the properties within the identified lenses. For the CTDs this was defined as the heterogenous  $\Theta$  layer. For the glider measurements this was defined as the heterogenous  $\Theta$  layer in the dive that contained the core  $\Theta$  minimum.

Author contributions. DDP conducted processing and analysis of glider measurements with input from DCEB, KJH and BYQ. GAL was responsible for the glider preparation, deployment and recovery. SL provided the pH sensor and delivered technical support. EH processed the pH sensor output and supported analysis. SES was responsible for processing the CTD measurements, Winkler titrations for O<sub>2</sub> and calculated the freshwater fractions. PLY was Chief-scientist of the 2010-11 expedition, Co-chief scientist of the NBP22-02 expedition, and collected and processed DIC measurements from 2010/2011 and 2022. FG processed and plotted seal tag measurements and contributed to interpretation alongside BGMW and KJH. DDP wrote the paper with input and revisions from all authors.

Competing interests. One author is a member of the editorial board of OS.

Acknowledgements. Funding from the Department of Science at the University of East Anglia supported DP. This work is from the TARSAN project, a component of the International Thwaites Glacier Collaboration (ITGC). Support from National Science Foundation (NSF: Grant 1738992) and Natural Environment Research Council (NERC: Grant NE/S006419/1). Logistics provided by NSF-U.S and NERC-British Antarctic Survey. Antarctic Program and NERC-British Antarctic Survey. ITGC Contribution No. ITGC-154. We also acknowledge funding from NERC (Grant: ARTEMIS NE/W007045/1) to KH and NSF (Grant: 1941483 - NSFGEO-NERC: ARTEMIS) to PLY. We thank the United States Antarctic Program (USAP), including the captain, crew, and science team onboard the Nathaniel B. Palmer (NBP) 22-02 for field support. We also thank the UEA glider group for piloting the gliders during this campaign and the Ocean Technology and Engineering Group at the National Oceanography Centre for providing the Lab-On-Chip pH sensor. Funding to UEA (NERC: Grant NE/J005703/1 and University of St Andrews (NERC: Grant NE/J005649/1) facilitated seal tag observations.

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
