# Peer review of "Cold lenses in the Amundsen Sea: Impacts of sea ice formation on subsurface pH and carbon"

_EGUsphere, 2025_

## Author Comment (AC1)

**Review of Pickup et al. "Cold lenses in the Amundsen Sea: Impacts of sea ice formation on subsurface pH and carbon" submitted to Ocean Science.**

**We are grateful to the reviewer for their helpful comments and suggestions. Our responses to their feedback are provided below, in bold. Any text amended and copied from the manuscript is also in blue.**

Brief summary

This study investigates subsurface "cold lenses" found beneath the Dotson Ice Shelf region of the Amundsen Sea polynya, Antarctica. These lenses are distinct pockets of cold ($\Theta < -1.7$ °C), salty, and dense water located at 240–500 m depth. High-resolution ocean glider measurements (temperature, salinity, dissolved oxygen, pH, and dissolved inorganic carbon, DIC) reveal that these lenses are colder, more saline, and denser than the overlying Winter Water (WW), but fresher and less dense than underlying modified Circumpolar Deep Water (mCDW). They exhibit slightly higher dissolved oxygen, lower pH, and elevated DIC, indicating intense surface cooling and brine rejection during sea ice formation. Two formation mechanisms are proposed: (1) formation in shallow coastal polynya regions (e.g., Martin Peninsula), where strong cooling and brine rejection drive dense water downslope to its neutral buoyancy depth (~400 m), and (2) local deep convection ("convective chimneys") during winter, with subsequent subsurface trapping. Seal tag data support high surface heat loss and deep mixed layer formation, suggesting sea ice production rates of ~3 cm/day.Ten lenses were identified, ranging from ~5 to 25 km in horizontal extent, and are likely recurring features formed annually. These lenses potentially reduce subsurface heat content, possibly limiting heat transport beneath ice shelves and acting as a barrier to basal melting if advected under the Dotson Ice Shelf. Furthermore, they may provide a mechanism for transporting carbon-rich water deeper than typical WW, influencing regional carbon budgets. Overall, the lenses highlight the importance of shallow shelf processes in shaping Antarctic subsurface water properties and carbon dynamics.

The paper is overall well written and presented. The topic is highly relevand and add to the few studies of ice-ocean interaction. Before I can recommend the paper for publication I would like the authors to take into account my specific comments below.

**Thank you for your positive comments on our work and helpful suggestions to strengthen it. We would just like to clarify that the study investigates subsurface cold lenses as captured by the glider and ship's CTD observations in the area north of the Dotson Ice Shelf region.**

Specific comments:

Figure 2: In order for the reader to easily follow what is shown, I suggest that you spell out the names of the water types e.g. AASW, mCDW, WW, Lens. Can be difficult to follow for scientists that is not local to the area. Suggest to change the depth bar so the cold water lences between 200-500 m has a distinct color. Then it will be easier to localise the cold lenses in the T-S space.

**We agree with the suggestion to spell out the names of the water masses, and have updated the figure accordingly so that AASW, mCDW and WW are written in full. Regarding the depth colourbar, we appreciate the suggestion to use a distinct colour to highlight the 200–500 m depth**

**range. However, not all water within this depth interval corresponds to the cold-water lenses, so a uniform change of colour along the depth axis could be misleading. Instead, we have added a boundary that delineates the lens properties directly in the T–S diagram. This makes the cold-water lenses easier to identify while ensuring that the depth colourbar continues to represent depth consistently.**

[Figure]

**Figure 2. Temperature-salinity diagram for measurements in the study area from the gliders and CTD casts coloured by depth. Local water masses are labelled and the freezing line was calculated using TEOS-10 (McDougall et al., 2010}. End members are shown with a cross and grey dashed lines indicate water mass mixing lines. Black sloping lines depict potential density (kg m³). The black dashed line depicts the location and properties of the lenses.**

You have indicated a mCDW-glacial meltwater mixing line. Could that be similar to the "Gade-line"? In a T–S diagram below an infinit ice cover, a melt line with an observed slope of 2.5 °C per salinity unit corresponds to the Gade slope.

Reference: Gade, H. G. Melting of ice in sea water: a primitive model with application to the Antarctic ice shelf and icebergs. J. Phys. Oceanogr. 9, 189–198 (1979).

**Yes, that is correct. Where it is introduced, we have now noted that the mCDW-glacial water mixing line can also be known as the Gade line and referenced Gade (1979) for clarity.**

Line 211: During sea ice formation, lighter oxygen isotopes are favoured in the ice and heavier isotopes remain in the water, lowering the δ18O of the water. This is not correct.

The fractionation effect during freezing is relatively small but tends to favour the 18O isotope in the ice compared to the residual liquid water. Sea ice typically has a δ18O value close to that of the source seawater, with a slight enrichment (more positive δ18O). In contrast, meteoric ice (ice formed from precipitation) is strongly depleted in 18O (more negative δ18O) compared to seawater. See

Moore et al. (2017) Fractionation of hydrogen and oxygen in artificial sea ice… Cold regions Science and Technology 142:93-99.

**We agree with the reviewer and have amended the text appropriately to ensure the interpretation of δ18O is correct (see response to comment on Line 207 – 245 to see this). As it is true that sea ice formation has a small effect on fractionation, we have rewritten elements of Section 3.3 that focus on δ18O. The point we want to make overall is that a sea ice formation signal is present within the lenses. Instead of relying on δ18O to do this, we will focus more on the freshwater fractions in Figure 6. Here there is a negative contribution from sea ice melt (suggesting sea ice formation) at the depth of the lenses, where there is also a decrease in δ18O.**

Line 219: The more negative δ18O within the lenses than in WW suggests more intense sea ice formation as the water has gotten more isotopically light. This is confusing. Do you mean:

The more depleted δ18O values within the lenses than in WW suggest a more intense sea ice formation (as the sea ice brine is more depleted in δ18O)…?

**We thank the reviewer for these clarifications (including the comment above and below), and we note that this section of the paper has been fully revised (see response to comment below).**

Line 207-245: I suggest to rewrite this section where you specify if an isotope is "enriched" or "depleted".

**We agree, this section has been amended to ensure the correct interpretation:**

**Lenses observed in each CTD cast are located at different depths, with the Θ minimum found within the range of 240 and 405 m (Fig. 5). The varying profile shapes within the lenses possibly reflect the same feature observed by the glider transects, where the difference in lens shape and size depends on which part of the lens was captured by the single profile. Three of the CTDs with lenses (Fig. 4) were additionally sampled for δ18O and DIC as shown in Fig. 5d and e. For CTD 8 the δ18O decreases from -0.45 ‰ at 200 m in WW to -0.50 ‰ at 300 m within the lens. For CTD 18 the decrease in δ18O is from -0.40 ‰ at 200 m (WW) to -0.48 ‰ within the lens at 400 m. For CTD 279 the decrease is from -0.47 ‰ at 245 m (WW) to -0.55 ‰ at 345 m, just below the lens. In the WW (depths between 100 - 200 m) δ18O is approximately 0.5 ‰ lower than in mCDW (depths greater than 450 m). The δ18O values of the lens cores are more negative to those of WW (Table 1). However, it is important to consider that there are only seven data points of δ18O within the lenses and the depth at which a measurement was taken may skew interpretation.**

**The effect of sea ice formation and melt is small on δ18O in relation to meteoric inputs. During sea ice formation, the heavier isotope, 18O, is favoured in the ice, and lighter 16O remains in seawater, lowering the 18O of the water affected by sea ice formation. To separate the effects of meteoric input and sea ice formation, the calculated freshwater fractions from δ18O are used to interpret the processes affecting the lenses. The freshwater fractions shown in Fig. 6 highlight a negative sea ice contribution to the lenses which indicates net sea ice formation. The most notable decrease is in CTD 18, with CTD 8 being less pronounced. There is no δ18O measurement at the depth within the lens from CTD 279, but the measurement just below the lens shows a decrease in the sea ice melt contribution. Water above the lenses, especially close to the surface, has a positive sea ice contribution, indicating net sea ice melt which has occurred locally in the spring.**

DIC concentrations in the lenses are slightly higher than in WW, by a maximum of 10 µmol kg−1 (Table 1). Similar to pH and O2, mCDW has the highest DIC concentration (2261 µmol kg−1) as it has spent the longest time not exchanging with the atmosphere. Additionally, DIC accumulates in the water column due to the breakdown of organic matter which produces DIC. The CTD casts with a lens are compared with a CTD profile where a lens was not observed (Fig. 5). For temperature, salinity, O2 and δ18O, the comparison CTD cast (CTD 194) was in the same location as the lens CTD cast (CTD 18), 20 days later (Fig. 4). No additional CTD casts were carried out in the region of the Dotson-Trough where DIC was sampled, so, two CTD casts from the ASPIRE campaign, which was carried out in December 2010 - January 2011 (Yager et al., 2012), within 25 km of CTD 18 were used for comparison (Fig. 4). Three consecutive dives made by SG579 after those within lens A were selected for a comparison of pH profiles in the lens with surrounding water.

Temperature and salinity are lower in the lenses than in nearby water at the same depth (Fig. 5) by a maximum of 1.50 ◦C and 0.16 g kg−1, respectively. The dissolved O2 concentration is higher in the lenses than in surrounding water and the δ18O in the lenses is approximately 0.18 ‰ lower than surrounding water, indicating a process that has occurred in the lens water mass, but not in nearby surrounding water. The lens has DIC concentrations very similar to one of the ASPIRE CTD casts (CTD 9) but approximately 18 µmol kg−1 lower than the other (CTD 72, collected about 2 weeks later in the season and over a deeper part of the trough). The lower DIC concentration is consistent with pH which is 0.02 higher in the lens than in surrounding water at the same depth.

Evidence of deep convection from the surface to depths less than 400 m or to the seabed (if shallower than 400 m) as indicated by seal profiles, were observed in the shallow shelf around Martin Peninsula and to the east of the Dotson-Getz Trough. There are also a few profiles in deeper water across the trough (Fig. 7a). Temperatures within these profiles reach a minimum of -1.85 ◦C at depths of approximately 300 - 400 m (Fig. 7b). These temperatures are associated with a SA greater than 34.25 g kg−1, mirroring the properties of the lenses.

Line 254: The mCDW contribution would also have a low O2 concentration, high δ18O, high DIC concentration and low pH content. This should be changed to: The mCDW contribution would also have a low O2 concentration, less depleted δ18O values, high DIC concentration and low pH content.

**Yes, thank you. This statement has been revised accordingly.**

Line 308: same comment as in line 207-245.

**This has been amended as well:** The lenses are colder, saltier and denser than overlying WW and are associated with a higher DIC concentration, lower pH and an enhanced fraction of net sea-ice formation (as determined from delta18O-derived freshwater fractions).

Figure 8: Very simple figure. Do not think it is needed. If decided to keep, I would suggest you to change the left horizontal arrow to follow more the bedrock and point out just above the "depth of lenses". You could also consider to add the information on 18O, DIC, O2 and pH to the conceptual figure.

**We have decided to keep the figure for clarity, but are grateful to the reviewer on points to improve it. Please see below for the amended version. We have included information on the properties of the lenses and how these differ from overlying WW.**

[Figure]

**Figure 8 - Schematic depicting the two possible formation processes of the lenses on the left and their properties. The numbers reference the formation theory with (1) formation in shallow water due to sea ice formation and spilling into deeper water and (2) local chimneys of convective mixing. The right depicts the properties of the overlying WW.**

---

## Author Response (AR1)

Review of Pickup et al. "Cold lenses in the Amundsen Sea: Impacts of sea ice formation on subsurface pH and carbon" submitted to Ocean Science.

We are sincerely grateful to both reviewers for their time, thoughtful comments and constructive suggestions, which have helped us to improve the clarity and quality of the manuscript.

Some additional edits were made by authors in the revised manuscript beyond the suggestions by the reviewers. These mostly comprise of minor wording corrections and sentence restructuring for clarity. Additionally, oxygen saturation has been included in Table 1, with information on how this was calculated added to Section 2.1 (Lines 88 – 90). The oxygen saturation helps us make our point that the lenses have a higher oxygen concentration that surrounding water due to increased solubility in colder waters (Line 248).

**Reviewer 1**

We are grateful to the reviewer for their helpful comments and suggestions. Our responses to their feedback are provided below, in bold. Any text amended and copied from the manuscript is also in blue.

**Brief summary**

This study investigates subsurface "cold lenses" found beneath the Dotson Ice Shelf region of the Amundsen Sea polynya, Antarctica. These lenses are distinct pockets of cold ( $\Theta

Figure 2. Temperature-salinity diagram for measurements in the study area from the gliders and CTD casts coloured by depth. Local water masses are labelled and the freezing line was calculated using TEOS-10 (McDougall et al., 2010). End members are shown with a cross and grey dashed lines indicate water mass mixing lines. Black sloping lines depict potential density (kg m³). The black dashed line depicts the location and properties of the lenses.

boundary that delineates the lens properties directly in the T–S diagram. This makes the coldwater lenses easier to identify while ensuring that the depth colourbar continues to represent depth consistently.

You have indicated a mCDW-glacial meltwater mixing line. Could that be similar to the "Gade-line"? In a T—S diagram below an infinit ice cover, a melt line with an observed slope of 2.5 °C per salinity unit corresponds to the Gade slope.

Reference: Gade, H. G. Melting of ice in sea water: a primitive model with application to the Antarctic ice shelf and icebergs. J. Phys. Oceanogr. 9, 189–198 (1979).

Yes, that is correct. Where it is introduced, we have now noted that the mCDW-glacial water mixing line can also be known as the Gade line and referenced Gade (1979) for clarity.

Line 211: During sea ice formation, lighter oxygen isotopes are favoured in the ice and heavier isotopes remain in the water, lowering the  $\delta 180$  of the water. This is not correct.

The fractionation effect during freezing is relatively small but tends to favour the 180 isotope in the ice compared to the residual liquid water. Sea ice typically has a  $\delta$ 180 value close to that of the source seawater, with a slight enrichment (more positive  $\delta$ 180). In contrast, meteoric ice (ice formed from precipitation) is strongly depleted in 180 (more negative  $\delta$ 180) compared to seawater. See Moore et al. (2017) Fractionation of hydrogen and oxygen in artificial sea ice... Cold regions Science and Technology 142:93-99.

We agree with the reviewer and have amended the text appropriately to ensure the interpretation of  $\delta 180$  is correct (see response to comment on Line 207-245 to see this). As it is true that sea ice formation has a small effect on fractionation, we have rewritten elements of Section 3.3 that focus on  $\delta 180$ . The point we want to make overall is that a sea ice formation signal is present within the lenses. Instead of relying on  $\delta 180$  to do this, we will focus more on the freshwater fractions in Figure 6. Here there is a negative contribution from sea ice melt (suggesting sea ice formation) at the depth of the lenses, where there is also a decrease in  $\delta 180$ .

Line 219: The more negative  $\delta$ 180 within the lenses than in WW suggests more intense sea ice formation as the water has gotten more isotopically light. This is confusing. Do you mean:

The more depleted  $\delta$ 180 values within the lenses than in WW suggest a more intense sea ice formation (as the sea ice brine is more depleted in  $\delta$ 180)...?

We thank the reviewer for these clarifications (including the comment above and below), and we note that this section of the paper has been fully revised (see response to comment below).

Line 207-245: I suggest to rewrite this section where you specify if an isotope is "enriched" or "depleted".

**We agree, this section has been amended to ensure the correct interpretation:**

Lenses observed in each CTD cast are located at different depths, with the  $\Theta$  minimum found within the range of 240 and 405 m (Fig. 5). The varying profile shapes within the lenses possibly reflect the same feature observed by the glider transects, where the difference in lens shape and size depends on which part of the lens was captured by the single profile. Three of the CTDs with lenses (Fig. 4) were additionally sampled for  $\delta$ 180 and DIC as shown in Fig. 5d and e. For CTD 8 the  $\delta$ 180 decreases from -0.45 ‰ at 200 m in WW to -0.50 ‰ at 300 m within the lens. For CTD 18 the decrease in  $\delta$ 180 is from -0.40 ‰ at 200 m (WW) to -0.48 ‰ within the lens at 400 m. For CTD 279 the decrease is from -0.47 ‰ at 245 m (WW) to -0.55 ‰ at 345 m, just below the lens. In the WW (depths between 100 - 200 m)  $\delta$ 180 is approximately 0.5 ‰ lower than in mCDW (depths greater than 450 m). The  $\delta$ 180 values of the lens cores are more negative to those of WW (Table 1).

However, it is important to consider that there are only seven data points of  $\delta$ 180 within the lenses and the depth at which a measurement was taken may skew interpretation.

The effect of sea ice formation and melt is small on  $\delta180$  in relation to meteoric inputs. During sea ice formation, the heavier isotope, 180, is favoured in the ice, and lighter 160 remains in seawater, lowering the 180 of the water affected by sea ice formation. To separate the effects of meteoric input and sea ice formation, the calculated freshwater fractions from  $\delta180$  are used to interpret the processes affecting the lenses. The freshwater fractions shown in Fig. 6 highlight a negative sea ice contribution to the lenses which indicates net sea ice formation. The most notable decrease is in CTD 18, with CTD 8 being less pronounced. There is no  $\delta180$  measurement at the depth within the lens from CTD 279, but the measurement just below the lens shows a decrease in the sea ice melt contribution. Water above the lenses, especially close to the surface, has a positive sea ice contribution, indicating net sea ice melt which has occurred locally in the spring.

DIC concentrations in the lenses are slightly higher than in WW, by a maximum of 10  $\mu$ mol kg–1 (Table 1). Similar to pH and O2, mCDW has the highest DIC concentration (2261  $\mu$ mol kg–1) as it has spent the longest time not exchanging with the atmosphere. Additionally, DIC accumulates in the water column due to the breakdown of organic matter which produces DIC. The CTD casts with a lens are compared with a CTD profile where a lens was not observed (Fig. 5). For temperature, salinity, O2 and  $\delta$ 18O, the comparison CTD cast (CTD 194) was in the same location as the lens CTD cast (CTD 18), 20 days later (Fig. 4). No additional CTD casts were carried out in the region of the Dotson-Trough where DIC was sampled, so, two CTD casts from the ASPIRE campaign, which was carried out in December 2010 - January 2011 (Yager et al., 2012), within 25 km of CTD 18 were used for comparison (Fig. 4). Three consecutive dives made by SG579 after those within lens A were selected for a comparison of pH profiles in the lens with surrounding water.

Temperature and salinity are lower in the lenses than in nearby water at the same depth (Fig. 5) by a maximum of 1.50  $^{\circ}$ C and 0.16 g kg–1, respectively. The dissolved O2 concentration is higher in the lenses than in surrounding water and the  $\delta$ 18O in the lenses is approximately 0.18 % lower than surrounding water, indicating a process that has occurred in the lens water mass, but not in nearby surrounding water. The lens has DIC concentrations very similar to one of the ASPIRE CTD casts (CTD 9) but approximately 18  $\mu$ mol kg–1 lower than the other (CTD 72, collected about 2 weeks later in the season and over a deeper part of the trough). The lower DIC concentration is consistent with pH which is 0.02 higher in the lens than in surrounding water at the same depth.

Evidence of deep convection from the surface to depths less than 400 m or to the seabed (if shallower than 400 m) as indicated by seal profiles, were observed in the shallow shelf around Martin Peninsula and to the east of the Dotson-Getz Trough. There are also a few profiles in deeper water across the trough (Fig. 7a). Temperatures within these profiles reach a minimum of -1.85 °C at depths of approximately 300 - 400 m (Fig. 7b). These temperatures are associated with a SA greater than 34.25 g kg-1, mirroring the properties of the lenses.

Line 254: The mCDW contribution would also have a low O2 concentration, high  $\delta$ 18O, high DIC concentration and low pH content. This should be changed to: The mCDW contribution would also have a low O2 concentration, less depleted  $\delta$ 18O values, high DIC concentration and low pH content.

Yes, thank you. This statement has been revised accordingly.

Line 308: same comment as in line 207-245.

This has been amended as well: The lenses are colder, saltier and denser than overlying WW and are associated with a higher DIC concentration, lower pH and an enhanced fraction of net sea-ice formation (as determined from delta180-derived freshwater fractions).

Figure 8: Very simple figure. Do not think it is needed. If decided to keep, I would suggest you to change the left horizontal arrow to follow more the bedrock and point out just above the "depth of lenses". You could also consider to add the information on 18O, DIC, O2 and pH to the conceptual figure.

We have decided to keep the figure for clarity, but are grateful to the reviewer on points to improve it. Please see below for the amended version. We have included information on the properties of the lenses and how these differ from overlying WW.

Figure 8 - Schematic depicting the two possible formation processes of the lenses on the left and their properties. The numbers reference the formation theory with (1) formation in shallow water due to sea ice formation and spilling into deeper water and (2) local chimneys of convective mixing. The right depicts the properties of the overlying WW.

**Reviewer 2**

We are grateful to the reviewer for their helpful comments and suggestions. Below we provide the reviewer's comments in plain black text and our responses to their feedback are provided in bold. Revised text to be added to the paper is shown in blue.

Overall this is a well written, scientifically rigorous study focused on observations of subsurface cold water lenses and a selection of their physical and biogeochemical signatures in the vicinity of the Dotson Ice shelf and the Dotson-Getz trough and surrounding environs. The observations presented here warrant prompt publication as they are incredibly difficult to capture and scientifically novel.

Thank you for your positive comments and appreciation of our novel data set.

Specific comments are below:

Introduction: There is a new Nature Reviews article on Antarctic coastal polynya's that may be worth referencing <a href="https://www.nature.com/articles/s43017-024-00634-x">https://www.nature.com/articles/s43017-024-00634-x</a>

Thank you for this helpful suggestion. This reference has been added in two places in the introduction:

Coastal polynyas are formed by katabatic winds that push sea ice away from the coast; the newly exposed surface water cools and refreezes before being blown offshore, continually generating open areas for new ice formation during winter (e.g. Golledge et al., 2025).

A recent review on Antarctic polynyas by Golledge et al. (2025) highlighted that many gaps in understanding processes in polynyas - including the role of long-term carbon sequestration in the Amundsen Sea- stem from limited observations.

A paper by Couto et al., 2017

https://agupubs.onlinelibrary.wiley.com/doi/full/10.1002/2017JC012840 similarly used gliders to track subsurface eddy features (with very different water mass characteristics) on the Western Antarctic Peninsula. It would be a nice reference to highlight and compare methodologies.

Thank you for bringing this paper to our attention and we agree. The reference and some discussion have been added to the introduction:

The Rossby radius in this region is less than 10 km (Chelton et al., 1998), so mesoscale features, such as eddies and meanders, are only a few kilometres in radius. Gliders have been used previous

on the West Antarctic Peninsula to detect eddies on the order of 10 km (Couto et al., 2017). With biogeochemical sensors, including a novel pH sensor, we identify the properties of observed mesoscale features in the Amundsen Sea, and discuss their formation.

We have also used this study as a way of supporting the fact that our glider profiles likely captured all of the lenses as they were 2-3x bigger than the eddies seen by Couto et al. 2017. This sentence is located towards the end of the discussion:

In our study, glider profiles were spaced 1–2 km apart, comparable to the spacing used by Couto et al. (2017), who successfully resolved eddy features with scales of 10 km. Given that the lenses we observed were up to 2–3 times larger than those eddies, we are confident that our glider resolution was sufficient to capture the full and representative distribution of lenses present in the study area.

**Methods:**

Line 72 - 80: How significant were the observed salinity spikes prior to removal? What approximate vertical resolutions were the data collected at at the 5s intervals? Was CTD data used from both downcast or upcast, or just the upcast data in conjunction with the slow upcast sampling for the pH sensor? I am surprised there's a thermal lag issue for such slow vertical speeds on upcasts for pH sampling and wonder if upcast only data is considered if the Garau method is necessary. How meaningful are the corrections to the final results of the paper?

Salinity spikes, were on the magnitude of up to  $\pm$  1 PSU. There is approximately a salinity measurement every 0.5-1 m with a 5s interval sampling time. The Garau method is included in the Seaglider processing toolbox used for this dataset (Queste et al., 2012, doi: 10.1109/AUV.2012.6380740), and we have added this reference for clarity:

Temperature and conductivity sensors (Sea-Bird CT Sail) were integrated on both Seagliders with sampling intervals of 5 seconds which corresponded to a measurement approximately every 0.5 - 1 m. The raw outputs were processed using the UEA Seaglider Toolbox (Queste et al., 2012) which incorporates corrections for the thermal lag of the un-pumped conductivity cell that produced artificial salinity spikes (on the scale of ± 1 PSU) following Garau et al. (2011).

For all variables, except for pH, both the upcast and downcast are used. We have added a sentence to clarify this as well as the speed of the descent:

For pH, only measurements taken during the ascent are reported in this study due to the nature of the glider flight which was programmed to travel relatively slowly during the ascent at a speed of 0.09 m s-1, yielding a higher vertical resolution of pH measurements. For all other sensors, both ascent and descent measurements are included.

The mean difference between the raw and final dataset of salinity is -0.0011 and the RMS difference is 0.0075. These values have been added to the methods.

The mean difference and root mean squared difference between the processed and raw versions of salinity measurements were -0.0011 and 0.0075, respectively.

Line 85: 3km is quite far and 6 profiles are not very many. I don't have an issue with the offset, but I suspect it would be helpful to say that this correction is small relative to the scale of the measured oxygen differences between the lens features of interest and surrounding waters.

We agree and have added this clarification:

This correction is small relative to the magnitude of the O2 differences observed between the lenses and the surrounding waters.

**Results:**

lines 140 - 142: I found this section confusing. I understand what you're going for referring to two temperature minima, but for the full dataset there's really only one minimum, consider rephrasing to clarify or highlighting that it is the minimum of temperature on either side of a salinity value in the first sentence. Furthermore, including a box or marker on Figure 2 of what 'minima' you are referring to would be helpful.

We agree and have amended the text to make it clearer. In Figure 2 we have outlined the range in  $\Theta/SA$  values that were detected inside the lenses to better distinguish those values from the  $\Theta/SA$  values of the overlying WW at the lower salinity

Analysis of the water masses in temperature-salinity space reveals two distinct salinity ranges where  $\Theta$  is less than -1.60 °C (Fig. 2) The more saline water mass (with SA greater than 34.2 g kg-1) is denser with temperatures closer to the freezing line (as calculated using TEOS-10 (McDougall et al., 2010). The less saline water mass (with SA less than 34.2 g kg-1) is the WW layer that lies below the AASW (Fig. 3a and b), with an SA between 34.05 and 34.15 g kg-1 (Fig. 2).

Line 142: You refer to the melting-freezing line here, but in the figure it only says 'freezing line.' I recommend consistency for clarity.

**This has been amended in the text.**

Figure 3: I recommend plotting the start and end locations on the map so it's easier to reference the figures on the left, which are in distance traveled. The light gray tracks are very difficult to make-out.

Markers for the start have been added to Figure 3c as purple crosses. The colour has been changed to blue to make the track clearer.

Discussion:

Line 264: The reference formatting looks incorrect.

Yes, there was a typo in the citation for Rysgaard et al., 2011. This has been correc